# Maize Internode Autofluorescence at the Macroscopic Scale: Image Representation and Principal Component Analysis of a Series of Large Multispectral Images

**DOI:** 10.3390/biom13071104

**Published:** 2023-07-11

**Authors:** Marie-Françoise Devaux, Mathias Corcel, Fabienne Guillon, Cécile Barron

**Affiliations:** 1UR1268 BIA, INRAE, 44316 Nantes, France; mathias.corcel@gmail.com (M.C.); fabienne.guillon@inrae.fr (F.G.); 2UMR IATE, Univ Montpellier, INRAE, Institut Agro, 34060 Montpellier, France; cecile.barron@inrae.fr

**Keywords:** multispectral image series, multispectral image analysis, large images, histology, plant tissues, fluorescence, PCA, statistical image comparison

## Abstract

A quantitative histology of maize stems is needed to study the role of tissue and of their chemical composition in plant development and in their end-use quality. In the present work, a new methodology is proposed to show and quantify the spatial variability of tissue composition in plant organs and to statistically compare different samples accounting for biological variability. Multispectral UV/visible autofluorescence imaging was used to acquire a macroscale image series based on the fluorescence of phenolic compounds in the cell wall. A series of 40 multispectral large images of a whole internode section taken from four maize inbred lines were compared. The series consisted of more than 1 billion pixels and 11 autofluorescence channels. Principal Component Analysis was adapted and named large PCA and score image montages at different scales were built. Large PCA score distributions were proposed as quantitative features to compare the inbred lines. Variations in the tissue fluorescence were clearly displayed in the score images. General intensity variations were identified. Rind vascular bundles were differentiated from other tissues due to their lignin fluorescence after visible excitation, while variations within the pith parenchyma were shown via UV fluorescence. They depended on the inbred line, as revealed by the first four large PCA score distributions. Autofluorescence macroscopy combined with an adapted analysis of a series of large images is promising for the investigation of the spatial heterogeneity of tissue composition between and within organ sections. The method is easy to implement and can be easily extended to other multi–hyperspectral imaging techniques. The score distributions enable a global comparison of the images and an analysis of the inbred lines’ effect. The interpretation of the tissue autofluorescence needs to be further investigated by using complementary spatially resolved techniques.

## 1. Introduction

Plant organs are composed of different tissues made up of cells whose size, shape, intracellular content, structure and cell wall composition vary according to the taxon and species to which they belong [1]. Within the same species, the relative proportion of tissues and the composition of their cell walls also vary. In maize, for example, the relative proportions of pith and rind, the size of parenchyma cells, the size and density of the vascular bundle and the relative quantity of lignified versus nonlignified tissues vary according to genotype [2,3,4,5], maturity [5,6,7,8,9,10], agroclimatic conditions [3,11] or cultural practice [12]. These variations affect plant life [12,13,14,15,16]. The structure and composition of plant biomass also influence its use for food and feed and for the production of biofuels or chemical compounds [3,7,17,18,19,20,21,22,23,24,25,26,27,28,29]. Therefore, the study and quantification of the heterogeneity and variability of tissue organisation and chemical composition are important. 

Imaging techniques providing a representative field of view and a spatial resolution adapted to the cell size scale are needed to consider the scales ranging from the organ to cell wall. As an example, maize stem diameters are approximately 1–2 cm^2^, while the cell diameter is approximately 60 µm [2]. In the case of maize stems, microscope slide scanners or macroscopes have been proposed to acquire images of whole stem sections [2,3]. The imaging method also needs to be able to quantify variations in tissue composition. Staining methods such as FASGA, Weisner or Maule staining have been used to locate and quantify lignin variations in lignocellulosic samples [30,31,32]. The interpretation is not straightforward, and staining is an added step in the sample preparation workflow that can have an effect on repeatability. Fourier transform infrared (FTIR) or Raman microspectroscopy are powerful techniques for mapping the distribution of cell wall polymers at the cellular or subcellular levels [33]. They have the advantage of not requiring any labelling. However, these techniques are time-consuming and only allow imaging of a small area of the sample. 

Alternatively, fluorescence imaging can be used. It has the advantage of requiring little tissue preparation, and there are imaging devices adapted to observe samples on a macroscopic scale [2,34]. In plant cell walls, lignin and hydroxycinnamic acids account for most of the autofluorescence and can be differentiated by their autofluorescence profiles [33,34,35,36,37,38,39,40,41]. Hydroxycinnamic acids emit blue fluorescence under UV excitation at approximately 350 nm [37,38], while lignin excited using UV and visible light emits blue, green and red fluorescence [35,39,40,41]. The relative proportions of phenolic compounds and their environment (pH, presence of quenching molecules, etc.) result in variable tissue fluorescence responses [36,38,40,42] that can be interpreted as a fluorescence tissue signature. Autofluorescence multispectral and hyperspectral imaging in the UV–visible range has been proven relevant for tracking the evolution of wheat grain outer tissues during development [43] or the relative distribution of lignin and hydroxycinnamic acids in the maize stem [2]. Ghaffari et al. (2019) [43] acquired hyperspectral autofluorescence images using the spectral mode in a confocal microscope and multi-excitation conditions. Berger et al. (2021) [2] applied full-field autofluorescence multispectral imaging at the macroscopic scale, which provided good spatial resolution (less than 3 µm per pixel), high acquisition speed and large fields of view. 

The analysis of autofluorescence multi- or hyperspectral images requires specific tools. The chemometric approach is generally retained. In this framework, the multi- or hyperspectral images are analysed as a set of spectra [33,44,45]. The basic method is Principal Component Analysis (PCA), which produces the loadings that provide the spectral interpretation and the scores that are represented as score images that describe the spatial variations. Multivariate Curve Resolution (MCR) is another common technique that aims to extract pure components and their concentrations from a set of spectra. Both techniques can be used for descriptive purposes.

In order to analyse a series of images, different approaches have been proposed. In the study by Berger et al. (2021) [2], a reduction in the number of spectra has been carried out in the following way. The 34 images of the series were individually segmented on the basis of morphological information and mean autofluorescence spectra were calculated for each segmented region. The mean spectra were then compared using Principal Component Analysis. In the Multivariate Curve Resolution framework, the analysis of several images is called multiset analysis [46,47]. The authors developed the concept of data augmentation, which consists of grouping together in a single table all the spectra from all images. In the study by Ghaffari et al. (2019) [43], 40 images were analysed together in this way, but the large number of pixels in the whole image series did not enable the consideration of the entire series in a single pass. In the study by Ghaffari et al. (2019) [43], each image was 512 × 512 pixels, resulting in more than 10 million pixels in the series, and in the study by Berger et al. (2021) [2], the 34 images were large images of approximately 4000 × 4000 pixels. Therefore, a problem of analysing a very large amount of data together is created and limitations in the usual software and algorithms are present because these programs were not designed to address such large amounts of data. In addition, a comparison of the images may be desired and not just a comparison of the individual pixels. The usual practice is to extract features from the images, which are then analysed to determine whether the images are different or not.

The aim of this work was to propose a methodology for processing the large volume of data needed to quantify the spatial variability of plant composition and statistically compare different samples, taking account of biological variability. A two-step approach is proposed. First, Principal Component Analysis was adapted to process a series of large multispectral images. In Principal Component Analysis, the core of the method is the calculation of the variance/covariance matrix common to all the spectra of the entire collection. This calculation can be decomposed and iteratively performed by considering the individual images [48,49]. In this way, the loadings are obtained from the contribution of all the pixels of the image series. The loadings are then used to compute the score images for each multispectral image in the series. The second step was to compute the score distributions for each score image. The score distributions can be compared to the grey-level histograms commonly used in image analysis, which show both pixel intensities and their proportions [50]. We propose to utilise the score distributions as characteristic features extracted from multispectral images.

Maize stem was selected to demonstrate the value of the method. Sections were prepared from the internodes under the ears of different stems of four maize inbred lines. Large images containing more than 4000 × 4000 pixels × 11 multispectral channels were acquired. Variations in tissue fluorescence and their location as well as the specific signature of the inbred line were examined.

## 2. Materials and Methods

### 2.1. Samples

#### 2.1.1. Maize Inbred Lines

Four inbred lines selected for their contrasting digestibility were grown in Arras (France) in 2018 [17]. Five plants per inbred line were harvested at the silage stage, and only the internode located under the main ear was sampled. They were stored in 70% ethanol/water (*v*/*v*).

#### 2.1.2. Internode Cross-Sections

For each internode, a one cm long segment was sampled in the middle of the internode, from which 150 µm thick cross-sections (called sections in the following) were cut in-air with a gsl1 microtome (Design and production: Lucchinetti, Schenkung Dapples, Zürich, Switzerland) [51]. Sections were stored in 70% ethanol at 4 °C for one year until image acquisition.

### 2.2. Image Acquisition

Two sections per internode were analysed. For Line M21, a third section was acquired for two internodes, and for Line M24, one internode was damaged and discarded. The total number of sections was 40. Prior to image acquisition, the sections were rehydrated in water overnight at 4 °C. Each section was placed between two round microscope cover glasses (35 mm) for observation. Multispectral autofluorescence images were acquired using the same autofluorescence macroscope described in [2].

#### 2.2.1. Multispectral Autofluorescence Imaging

The multispectral autofluorescence images were acquired using the Multizoom AZ100M macroscope (Nikon France, Champigny sur Marne, France). The macroscope was equipped with a Q Imaging EXI Aqua monochrome camera with a 14-bit dynamic and an RGB-HM-S-IR filter wheel, which allowed for RGB colour image acquisition and grey-level intensities coded using 16,384 values. Excitation light was generated using an INTENSILIGHT mercury lamp (C-HGFI/C-HGFIE Precentred Fibre Illuminator, Nikon France, Champigny sur Marne, France). Four fluorescence filter cubes were used: UV1 (U1), UV2 (U2), BLUE (BL) and GREEN (GR), each with specific spectral specifications outlined in Table 1. To subtract the background, shading correction was applied to each image during acquisition. The total magnification was set to X4 by combining the AZ-Plan Fluor 2× lens (NA: 0.2/WD: 45 mm) and an ×2 optical zoom. With these settings, each image represented a field of view measuring 3.9 × 2.9 mm^2^ with a pixel size of 2.78 μm. The macroscope featured a Prior Proscan II (Nikon France, Champigny sur Marne, France) motorised stage, enabling large image acquisition. The NIS-Elements software (AR 5.02.02, Nikon France, Champigny sur Marne, France) with the ND-acquisition module was used to drive the macroscope for automated acquisition of large multispectral images. The multispectral sequence involved acquiring the four RGB images of the fluorescence filters in succession for a given field of view, starting with GREEN, followed by BLUE, UV2 and UV1 filter cubes. Exposure times were adjusted after viewing a few samples, as indicated in Table 1. 

#### 2.2.2. Multispectral Images and Pseudospectra

Multispectral images

The multispectral images contained 12 channels by merging the four RGB images as proposed in [34]. The procedure is repeated here. The channels were arranged in order from high to low wavelengths: blue, red and green channels of each RGB image acquired with filters UV1, UV2, BLUE and GREEN. The channels were named U1b, U1g, U1r, U2b, U2g, U2r, BLb, BLg, BLr, GRb, GRg and GRr. Channel U1r was removed from the sequence because it contained unwanted reflection from the excitation Rayleigh band. Therefore, the final multispectral image contained 11 channels (Figure 1).

Pseudospectra

The collection of 11 fluorescence intensities measured for individual pixels was referred to as pseudospectra [34]. As no photons were emitted at wavelengths higher than the excitation wavelength, the signals in channels BLb, GRb and GRg were absent for the visible filters (BLUE and GREEN). However, these channels were retained in the pseudospectra as a baseline reference. The pseudospectra can be divided into two regions: UV-induced fluorescence and visible-induced fluorescence, corresponding to the fluorescence obtained after UV excitation and BLUE/GREEN excitations, respectively.

### 2.3. Image Processing

All image processing was performed in the MATLAB 2022a environment (The Mathworks, Natick, MA, USA) using the image processing toolbox, the statistics and machine learning toolbox, and dedicated custom functions and scripts developed for a series of macrofluorescence multispectral images.

#### 2.3.1. Image Representation

Looking at images is only possible for monochrome and colour images. Multispectral images contain more than 3 channels and require appropriate representation tools to visualise them before they can be subjected to multivariate analysis. In this work, *RGB composite images of macrofluorescence* were calculated. Comparing several images creates additional problems. Common min and max values are necessary to compare intensities between images. Finally, in the case of large images, it is difficult to have a global view of several images and to be able to describe variations on the scale of details or regions in the images. In this work, automatic zoom region selections and image montages were set up to compare image series.

-RGB composite images of macrofluorescence

The *RGB composite image of macrofluorescence* was computed as described in [2]: The red channel of the RGB composite image was computed as the average of the red channels U2r, BLr and GRr. The green channel was computed as the average of the green channels U1g, U2g and BLg. The blue channel was computed as the average of blue channels U1b and U2b.

-Determining a common greyscale

To compare the intensity of images from one inbred line to the other, a MATLAB function specific to the current project was developed. First, the min and max values and the 1st and 99th percentiles of the greyscale histogram were calculated for each *RGB composite image of macrofluorescence*. From these values, the user is invited to define a common *min* and *max* value to represent images. The common colour scale was ensured by applying the MATLAB *imadjust* function with the *min* and *max* values and a *gamma* value chosen to better visualise the low fluorescence intensities. In the present study, the values were set to *min* = 0, *max* = 11,000 and *gamma* = 0.65.

-From series to detail: a multiscale representation

MATLAB functions have been specifically written for the project to set up a multiscale visualisation approach from the image series to the details. For each image, the zoomed regions were automatically selected in the centre and on the border of the sections. Montages of the whole series of images were automatically generated, either from the whole images or from the zoomed regions.

#### 2.3.2. Regions of Interest of the Internode Sections

The pixels corresponding to the internode section were identified from the background by considering the grey-level image computed as the sum of the 11 channels of the multispectral image and called the *sum of intensity image*. The *sum of intensity image* was segmented by thresholding after filling holes using the Otsu threshold (graythresh.m Matlab function) [50]. The threshold could have been adjusted after user checking. The threshold region was postprocessed via morphological opening and closing [52], using disks as structuring elements with sizes of 101 and 51 pixels, respectively.

#### 2.3.3. Principal Component Analysis of a Series of Large Multispectral Images

Principal Component Analysis is the basic method used in multispectral image analysis [44,45]. It considers the images as a set of pixels, i.e., a set of pseudospectra. The spatial nature of the multispectral image is initially disregarded. The loadings and scores are computed. The loadings provide the spectral interpretation, and the scores are refolded to be visualised as score images. They show the regions of the images where the pixels contrasted by the components are located.

In the present work, each large multispectral image contained between 17 and 41 million pixels encoded in 16 bits, and the 40 images analysed together represented almost 24 Go of data. To compare all images with each other in a single analysis, the Principal Component Analysis algorithm was adapted [48,49] and named *large PCA*.

In the large PCA software, all images are iteratively considered (Figure 2). Each multispectral image is unfolded in the form of a typical data table **X*_i_*** from which local variables are computed as follows:
-the local number of pixels *n_i_* contained in the data table **X*_i_***,-the local sum of all pseudospectra ***s_i_***:
(1)$$s_{i}=\sum_{j=1}^{n_{i}}{x_{i}}_{j}$$

where ${x_{i}}_{j}$ is the *j*th pseudospectrum of multispectral image **X*_i_***,
-the local contribution to the variance/covariance matrix is as follows:
(2)$$V_{i}={X_{i}}^{\prime }X_{i}$$
where **X*_i_*′** denotes the transpose of **X*_i_***.


The local variables are used to iteratively compute *n*, the total number of pixels included in the analysis, the global average pseudospectra and the global variance/covariance matrix, as shown in the equation of Figure 2. The variance/covariance matrix is subjected to singular value decomposition to obtain the eigenvalues and the loadings **L** of the large analysis.

The scores are computed by iteratively loading each unfolded multispectral image. The image is centred by subtracting the global average pseudospectrum and coded ${X_{c}}_{i}$, and the scores are obtained using the common loadings **L**, as follows:(3)$$C_{i}={X_{c}}_{i}L$$

The scores are finally refolded to form the score images.

In the present work, large PCA was applied to the sets of pixels included in the *regions of interest* of the sections.

#### 2.3.4. Representation of Large PCA Scores 

After large PCA, scores were saved on a disk in single precision. A representation has to be set up for score display and to compare score images from one image to the other. For this purpose, score intensities were coded to 8 bits, considering a common minimum and maximum for all images. The minimum and maximum values were a priori determined from the eigenvalue *λ*(*i*) of the large PCA component. The eigenvalues were used as an estimate of the standard deviation. The minima were set to −5 × *λ*(*i*) and the maxima to +5 × *λ*(*i*). The minima and maxima were used to linearly transform the scores to unsigned integers between 0 and 225.

#### 2.3.5. Distribution of Large PCA Scores

Displaying large PCA scores as images on a common scale allows qualitative comparison of different sections. Here, we propose to calculate, for each component, the observed distribution of pixel scores, which can be used as quantitative features to compare the multispectral images. The distributions were computed in two steps as follows:Step 1, during large PCA score computing:
-A large number of bins is a priori fixed with linear edges between each bin. The minimum and maximum edges are determined by considering the eigenvalue *λ*(*i*) of each component. A multiplicative factor is applied to the eigenvalue large enough to a priori include a maximum number of pixels with extreme scores. The minima were a priori set to −25 × *λ*(*i*)*,* the maxima to *+*25 × *λ*(*i*) and the number of bins to 10,000.-For each multispectral image, scores are computed as described in Section 2.3.3, and the score distributions are calculated immediately afterwards. The minimum and maximum edges can be updated if the minimum or maximum scores exceed the a priori values.-The series of distributions is saved at the end of the image series processing.
Step 2, postprocessing of the score distributions:
-For each large PCA component, the *total score distribution* is calculated by summing up all individual image score distributions. The cumulative distribution, called *the total cumulative score distribution*, is computed.-The percentiles of the *total cumulative score distribution* are next calculated. Here, percentiles corresponding to 0 to 100% of the pixels per step of 1% were considered.-The percentiles defined new edges, leading to bins with variable widths.-Finally, the new edges were used to interpolate the cumulative score distribution for each image and to calculate the corresponding score distribution.



### 2.4. Analysis of the Large PCA Score Distributions

To study the differences between the four inbred lines, score distributions were compared by Principal Component Analysis. Custom functions were run in the MATLAB 2022a environment (The Mathworks, Natick, MA, USA) using the singular value decomposition *svd* function. The resulting Principal Components were called *score distribution Principal Components*. The suitability of the score distributions to discriminate the four lines was evaluated via variance analysis of the score distribution Principal Components using the *anova1* MATLAB function.

## 3. Results

### 3.1. Exploring the Variability through Image Series Representation

#### 3.1.1. Montage of Image Series

In the present work, multispectral large images were acquired for two or three sections per internode. Montages of the entire series of *RGB composite images of macrofluorescence* are shown in Appendix A for the full sections and the two zoomed regions. These representations are useful for evaluating between image variations. After checking in Appendix A that the variations between the sections of the same internode were not visually large, an image montage was performed using one image per internode. This enables the comparison of the four lines on a single sheet (Figure 3).

With this representation, the following deductions were drawn: the lines differed by the section area and shape. M23 had the largest sections, while M21 and M24 had the smallest. The section area varied more for Lines M23 and M24 than for Lines M21 and M22. The section areas were measured as 1.54 ± 0.08 cm^2^, 2.10 ± 0.06 cm^2^, 2.65 ± 0.18 cm^2^ and 1.81 ± 0.16 cm^2^ for Lines M21, M22, M23 and M24, respectively. Line M23 sections were more elongated and those of M22 were larger.

In all sections, the fluorescence of the rind was different from that of the pith, indicating the presence of many lignified vascular bundles in this region [1]. Lignin fluorescence was observed after excitation in a large range of wavelengths with specific and strong fluorescence after visible excitation [2,35], leading to a broad range of emissions, including blue, green and red. Lignin fluorescence colour is known to vary according to its physical structure and environment (presence of quenching molecules, pH, mounting medium) [40,41,53]. In the four lines of the study, the rind colour was mainly pink/orange. The dots within the parenchyma were the lignified vascular bundles. The blue fluorescence of the parenchyma cell walls confirmed the strong fluorescence of hydroxycinnamic acids, which mainly emitted blue light under UV excitation at a neutral pH [2,35]. The fluorescence of the rind was more intense for Line M23 than for the other lines; that of M22 appeared weaker. M23 rind fluorescence also depended on the sections. From Appendix A, this was attributed both to variations between internodes and to possible variation in the section thickness.

#### 3.1.2. Example Images

Since no major differences in the overall structure and fluorescence properties were highlighted between images within each line, one image per line was retained. Two zoomed areas were selected to consider the two main anatomical structures evidenced: the pith and the rind (Figure 4). Their field of view of 4.17 × 4.17 mm^2^ was a compromise to show enough detail while maintaining visualisation of possible variability. The zoomed rind images showed that the rind fluorescence of Line M23 was caused by the strong pink fluorescence of the sclerenchyma sheath of the vascular bundles and the pink fluorescence of the rind parenchyma. The rind fluorescence observed for Line 21 was weaker and more orange. In Lines M22 and M24, the fluorescence of the rind parenchyma was largely blue. The fluorescence colour of the rind vascular bundles was clearly orange for Line M24. The overall fluorescence was weaker for M22.

In the centre of the sections, the blue fluorescence was nearly similar for all the lines. The blue fluorescence in the parenchyma cell walls depended on the relative proportions of ferulic and para-coumaric acids or on a variable amount of lignin in the parenchyma cell walls [23]. The zoomed images also showed that the relative number of vascular bundles varied; they were more numerous in M21. Their lignification was also probably different, as evidenced by the fluorescence of the sclerenchyma sheaths, which seemed more intense and orange in Line M21.

### 3.2. Average Pseudospectra

The mean pseudospectra of each multispectral image were computed and averaged over the inbred lines (Figure 5). Line M23 was the line with the most intense fluorescence, and M22 was the line with least fluorescence, especially in the visible-induced fluorescence. A strong red emission was specifically observed for Line M23, regardless of the excitation: refer to the U2r, BLr and GRr intensities. The shape of the mean pseudospectra varied according to the line mainly in the visible excitation range. In the UV excitation range, the blue emission was similar between the lines. Relative variations between green and red emissions after BLUE excitation were observed; compared to the red emission, a relative higher green emission (BLg) was observed for Lines M21 and M24 (ratio BLg/BLr of 0.60 ± 0.02 and 0.60 ± 0.04, respectively) than that for Lines M22 and M23 (ratio of 0.52 ± 0.05 and 0.46 ± 0.04, respectively). 

Based on the mean pseudospectra, a line effect can be observed via Principal Component Analysis. The results are shown in Appendix A but are not commented on here because this analysis does not allow the different causes to be differentiated: tissue-independent specific fluorescence behaviour, differences in the proportion of tissues with common fluorescence, and tissue-specific fluorescence. For a deeper analysis, we performed a large Principal Component Analysis on all the pixels included in the 40 images of the series.

### 3.3. Large Principal Component Analysis: loadings

Height components were computed describing 80.7, 16.1, 1.4, 1.0, 0.4, 0.2, 0.1 and 0.1% of the total variance. The loadings showed the main fluorescence variations. The first four are described here (Figure 6).

The loading 1 values were all negative, indicating a variation in overall intensity between pixels. This was mainly found in the fluorescence observed after BLUE and GREEN excitations. This loading was expected since the pixels within cells were considered together with cell wall pixels, and no normalisation between pixels or images was applied. The second loading highlighted variations between UV- and visible-induced fluorescence. This loading was expected to describe the differences between the lignified cell wall in the rind and vascular bundles and the blue parenchyma cell walls. The third loading mainly described the green fluorescence emission after BLUE excitation, as opposed to the red emission after UV2 and GREEN excitations. Fluorescence emission after BLUE excitation was expected to be characteristic of lignin fluorescence, and red emission was only observed for Line M23 (Figure 5). The fourth loading mainly described the differences in fluorescence emission after UV1 and UV2 emissions. The two UV filters, although they were close in terms of excitation and emission (see Table 1), were hypothesised to be able to distinguish between ferulic and para-coumaric acids [54,55]. This will be further discussed in 3.7.

In the following, we examined these four components through the visual representation of image scores and the quantification of differences between lines through the extraction of the score distributions.

### 3.4. Large Principal Component 1: General Intensity Variations

Figure 7 shows the Principal Component Score images for one section per internode and for the zoomed regions of the four example images. Please refer to Figure 3 and Figure 4 to identify regions and cell types. The montage for the whole series is found in Appendix A. In these images, the pixels with the strongest fluorescence are dark, as shown in loading one (Figure 6). The score images show that the strongest fluorescence was observed for Line M23, mainly in the rind. The lowest fluorescence was observed for Line M22, regardless of the tissue. The fluorescence of the vascular bundles was strong enough to clearly distinguish them in the whole section images.

From the zoomed images, the strongest fluorescence came from the sclerenchyma sheath of the vascular bundles, corresponding to the most lignified tissue in both the rind and the pith. The thickness of the rind bundle sheaths was larger for Line M23 than for the other lines. The rind parenchyma was also more fluorescent for Line M23.

The score distributions were computed as described in Section 2.3.4, and the average distribution per line is given in Figure 8. These distributions clearly differed in the case of the M23 and M22 lines. For all distributions, several modes were observed, with varying score values. The first mode corresponded to the lignified tissues, rind and vascular bundles (noted ‘L’ on the figure), while the second mode corresponded to the pith parenchyma cell walls (noted ‘p’). Finally, the largest scores accounted for the pixels in the cell lumen. The scores of the modes quantified the intensity of fluorescence. Thus, the fluorescence of lignified tissue could be ranked from highest to lowest in the order M23, M21, M24 and M22. The same trend was observed for the fluorescence in the pith parenchyma but with smaller differences between the lines.

Principal Component Analysis was applied to the score distributions. The first two Components accounting for 55 and 28% of the total variance showed the differences between the lines and the heterogeneity within inbred lines, i.e., the variability between internodes and sections. In the case of Line M21, one section showed more intense fluorescence than the other (the first section in Appendix A) due to a variation in the thickness of the section in this case. Analysis of the variance was applied to the Principal Components of the distribution to determine the significance of the differences between the inbred lines (Table 2).

Component 1 significantly discriminated Line M22 as having the most-numerous pixels with positive score values in the large PCA, i.e., the less-fluorescent internode sections. Component 2 significantly discriminated Line M23, followed by M22, from the two other lines. Line M23 was found to be specific due to its strong negative large PCA scores and Line M22 was specific due to its positive scores. Lines M21 and M24 were found to be similar with medium–large PCA scores.

The interpretation tools proposed in conjunction with the large PCA, i.e., score image representation and score distribution analysis were found to be relevant for comparing the series of multispectral images, detecting line differences and indicating the tissue origin of these differences. In the following, we propose to start with the analysis of the score distribution to obtain an immediate overview of the variability within the images. We will then move on to the spatial interpretation by examining representative score images. Principal components of score distributions are listed in Table 2, which shows their significance in describing the line effect.

### 3.5. Large Principal Component 2: UV and Visible Fluorescence

From Table 2, large PCA Component 2 appears to be the one that least describes a line effect based on its overall significance of the Principal Components of the score distribution. Principal Component 1 of the score distribution and loadings one, two and three are shown in Appendix A. The average score distributions and their Principal Components 2–3 are shown in Figure 9. The significant differences for Components 2 and 3 were caused by Lines M22, M23 and M24. Line M21 score distributions were too heterogeneous to be identified either differently or clearly similar to the other lines. The second large PCA Component described relative intensity variations between UV- and visible-induced fluorescence, mainly BLr and GRr channels (Figure 6). The score distributions showed that the red fluorescence of vascular bundles was high for Line M23, low for Line M22 and intermediate for Lines M21 and M24.

Three groups of pixel scores were distinguished in the average score distributions (Figure 9): negative scores, positive scores over 1500/2000 and intermediate scores. The zoomed areas of the example images are shown in Figure 9 (see Appendix A for the montage of the entire series of images). For the four lines, the images showed that the pixels with positive scores corresponded to the internode epidermis and the sclerenchyma sheaths of the rind vascular bundles with varying intensities depending on the line: over 3000 for M23 and between 2000 and 5000 for M21 and M24. In the case of Line M22, the scores were not clearly distinguished from the intermediate scores of the cell lumens. Pixels with negative scores corresponded to the parenchyma cell walls in the pith and the rind and to the pith vascular bundles. For Lines M22, M23 and M24, the vascular bundles in the pith were completely dark, i.e., showing a relatively higher UV-induced fluorescence intensity. In the case of Line M21, the sheaths on the top and bottoms of the vascular bundles in the pith showed positive large PCA scores, indicative of a relatively higher visible-induced fluorescence; this corresponded to the orange fluorescence observed in the RGB composite images of fluorescence in Section 3.1.2.

In conclusion, the second large Principal Components contrasted the epidermis and the vascular bundles of the rind from the bundles in the pith and from the parenchyma cell walls. In the rind, vascular bundles emitted a more intense fluorescence in red after visible excitation, and this relative intensity depended on the line.

### 3.6. Large Principal Component 3: BLUE Excitation and UV–GREEN-Induced Red Emission

Large Principal Component 3 showed a specific green emission after BLUE excitation that was negatively correlated with a red emission after GREEN and UV2 excitations. This was found to be partly correlated with the blue–green emission after UV2 excitation (Figure 6). Principal Components showed significant variations according to the inbred line (Table 2). The average score distributions are shown in Figure 10. Principal Component 1 of the score distribution discriminated Line M23 from the other lines (see Appendix A). The scatter plot of Principal Components 2 and 3 (Figure 10) showed a contrasted line effect for the three other lines.

Line M23 was the line with the highest number of pixels with positive large PCA scores 3, i.e., a high red emission after GREEN and UV2 excitation. From the score images in Figure 10 and Appendix A, these pixels were mainly located in the cell walls of the parenchyma near the rind, the rind parenchyma, especially just under the epidermis, and around the vascular bundles. Line M24 also contained pixels with positive large PCA scores, although the mode score of approximately 750 was lower than that of Line M23. These pixels were also located in the parenchyma near the rind and in the rind parenchyma; however, they were barely found near the vascular bundles. For Lines M21 and M22, bright pixels were also found in the parenchyma near the rind and around vascular bundles. The scores were smaller (<500) than the two other lines, as shown by the score distributions. Line M21 could be distinguished by Principal Components 2 and 3 of the score distributions. For this line, a relatively high proportion of pixels was found to have strong green fluorescence after BLUE excitation, i.e., negative large PCA scores: dark pixels in the score images. These pixels were mainly located in the vascular bundles of the rind and the pith but also in the rind parenchyma. Line M22 was specific for having an intermediate fluorescence behaviour with none of the extremely large PCA scores 3.

Two types of parenchyma fluorescence were shown by the third large PCA component: fluorescence of the parenchyma near the rind and the pith vascular bundles and fluorescence of the pith parenchyma. These two types of parenchyma were known to have different behaviours towards enzymatic degradation [7,24,55]. The parenchyma near the rind and around the vascular bundle in the pith were characterised by a relatively higher red emission after UV2 and green excitation and a low green emission after BLUE excitation. This was found to be particularly intense for Line M23.

### 3.7. Large Principal Component 4: UV1 and UV2 Fluorescence Behaviour

The fourth large PCA Component mainly described the relative fluorescence intensity variations between the two UV filters. Excitation covered the range 327–353 nm and 325–375 nm for UV1 and UV2 filters with emission over 380 nm and 420 nm, respectively. Ferulic and para-coumaric acids were found in maize stems [4,6,8]. Their fluorescence properties were similar but were reported to slightly differ after 365 nm excitation [54,56]. At neutral pH, a more specific blue fluorescence emission could be observed for ferulic acid and greener emission in the case of para-coumaric acid [54]. The differences between the UV fluorescence of lignin and hydroxycinnamic acids were also expected [33,35].

From Table 2, the first two principal components of the large PCA score distributions were highly significant for the inbred line effect. From Figure 11, Line M21 showed a large number of pixels with positive large PCA scores, i.e., a relatively higher UV1-induced fluorescence compared to Line M23, which showed a large number of pixels with negative scores, i.e., a relatively higher UV2-induced fluorescence. Lines M22 and M24 were intermediate.

Figure 12 and Appendix A show the score images. Regarding the four lines, the fourth large PCA Component generally contrasted the parenchyma near the rind as having a relatively higher fluorescence after UV1 excitation compared to the pith parenchyma. This fluorescence behaviour was consistently observed for the parenchyma near the vascular bundles. Despite some variability within the lines, the line effect was clearly visible, as shown in Figure 12 and Appendix A, and verified by the analysis of the score distributions. The image montage of the whole sections clearly showed large areas in Line M21 corresponding to the UV1 fluorescent parenchyma. The surface of these areas also depended on the internode, as shown by the section repetitions in Appendix A. Intensity variations were observed, with more contrasted bright and dark pixels in the case of Line M23 compared to Lines M22 or M21. The surface of the areas corresponding to the parenchyma near the vascular bundles also depended on the line. In the case of Line M23, at least two cell layers showed bright pixels, while in the case of Lines M24 or M22, one cell layer was observed.

The parenchyma regions near the rind and near the vascular bundles corresponded to those also observed in the case of large PCA score 3. However, for this 4th large PCA score, high large PCA scores were also observed for the sclerenchyma sheath of the vascular bundles in the rind. For these lines, the intensity varied according to the line and was very high for Line M23 and low for Line M22. In contrast, the rind parenchyma pixels were dark for all lines, showing a higher relative fluorescence after UV2 excitation.

Finally, in the pith, the vascular bundle fluorescence also depended on the line. They showed a higher UV2 fluorescence for Lines M22 and M23, i.e., darker bundles, compared to Lines M21 and M24. In the case of Line M23 and partly Line M22, some regions with dark pixels were highlighted in the rind. This was observed for two internodes.

## 4. Discussion

### 4.1. Multispectral Autofluorescence Imaging to Evaluate the Distribution of Biochemical Compounds of a Large Sample

Plants are known to be composed of different tissues at different scales whose chemical composition depends on the stage of development, the line or genotype, the growing conditions, the location in the stem, etc. Depending on the design of the experiment, it is not always clear in which tissue a chemical change will take place and whether this change will also occur in other tissues. These considerations justify the need to define tools and approaches to evaluate the spatial variability of composition in large samples, without any *a priori* information. Multispectral autofluorescence imaging can be used to assess the distribution of phenolic compounds of the plant cell wall [33,35]. Although not all compounds can be studied in this way, the advantage is that a series of samples can be analysed at the macroscopic scale. A comparison of the multispectral images is then needed based on their spectral and spatial content.

### 4.2. Representation and Principal Component Analysis of Multispectral Autofluorescence Images

The multispectral autofluorescence images consisted of 11 fluorescence channels. Although the number of channels was small, the multispectral images could not be directly displayed. The spectral dimension needed to be decreased to at least three or one to obtain a colour or monochrome image. The RGB composite autofluorescence images were proposed as a quick view, enhancing the UV-induced fluorescence in the blue channel and mostly the visible-induced fluorescence in the red and green channels. This representation was not sufficient to determine all observed relative variations. Principal Component Analysis was shown to be efficient in analysing the autofluorescence pseudospectra [2,34,48]. Principal Component Analysis has the advantage of grouping the correlated information in the first components and highlighting the noncorrelated information in the following components. Thus, the complementary information provided by the UV1 and UV2 filters could be shown in Components 3 and 4.

### 4.3. Principal Component Analysis Was Easy to Adapt to a Series of Images

Principal Component Analysis was applied to a series of autofluorescence images of maize internode sections with areas ranging between 1.5 and 2.67 cm^2^ and a pixel size below 3 µm. The number of pixels was very large: approximately 20 million for each image and more than 1 billion in total. Principal Component Analysis was adapted to address this issue and applied to the large multispectral image series. Calculating the variance–covariance matrix could easily be performed iteratively by successively loading each multispectral image. A parallelisation could be very easy to implement for very fast calculations. In the present case, extracting the loadings from the 11 × 11 large variance–covariance matrix of the 11 autofluorescence channels was straightforward.

The loadings and therefore the large Principal Components considered all variability contributed by all pixels in the image series. As a result, the percentages of variance of the Principal Components were affected by the large number of pixels. For example, here, the occurrence in all images of both pixels with high and low fluorescence signals represented more than 80% of the total variance. On the other hand, a Component with a low weight could represent significant variations for a relatively small number of pixels. This needed to be verified by examining the score images. Again, 1.4 and 1.0% of the variance of Components 3 and 4 were highly significant, as the Components showed different fluorescence behaviours of parenchyma tissues. To compare the pixel scores and more generally, multispectral images with each other, the challenge was to define additional interpretation tools.

### 4.4. Representing Large PCA Score Images of Large Multispectral Image Series

We have proposed a multiscale display of the image series, which enables the comparison of intensities from one image to another. The montage of whole images enables a rapid evaluation of the variability within the series and the selection of representative images. The montage of the zoomed regions has the advantage of allowing the comparison of the series and highlighting details such as vascular bundles or tissues inside vascular bundles. However, the visual comparison of the images remains subjective.

### 4.5. Score Distributions Were Efficient in Comparing Images

Another way to explore the information provided by the large PCA scores was to compute score distributions. For each large PCA component, a total score distribution was computed that defined the percentiles of the entire pixel collection. The score distribution was then recalculated for each multispectral image using the percentiles. In this way, it became possible to determine whether pixels with contrasting fluorescence behaviour were from specific images or from specific tissues. For example, in our study, for large PCA Components 1, 3 and 4, the fluorescence variations could be attributed to specific tissues with a clear line effect; this was the case for the general fluorescence intensity mainly observed for the rind and vascular bundles. This was also the case for the parenchyma near the rind and the vascular bundles; the cortical parenchyma and the sclerenchyma sheath of the vascular bundles in the rind showed specific UV fluorescence as attested by the large PCA Components 4 and 3. Additionally, the relative UV- or visible-induced fluorescence was deduced to be predominantly a tissue effect, regardless of the line.

Score distributions were an easy way to extract global quantitative features from multispectral large images. These distributions considered the tissue proportion in the section together with their spectral properties. Thus, a section with a small area would naturally show a higher proportion of rind compared to a section with a large area. This needed to be considered when drawing conclusions. In any case, the score distributions were found to be relevant for quickly comparing images with a large number of pixels, enabling a statistical comparison of sections within the same internode and internodes within the same line and demonstrating a line effect.

### 4.6. Maize Stem Tissues Can Be Differentiated by Their Autofluorescence Properties

Multispectral autofluorescence imaging at the macroscopic scale combined with image analysis and a statistical approach showed tissue-specific fluorescence to a very fine level, e.g., vascular bundle sheath for different inbred lines and without segmentation. We identified general variations in intensity with a dominant line effect. Fluorescence variations after UV and visible excitation enabled the discrimination of the rind vascular bundles from other tissues. Notably, variations within the pith parenchyma according to location were shown via UV fluorescence. These variations depended on the inbred line. Other authors have also found a diversity in cell wall local composition depending on tissue and localisation in maize stem sections. Corcel et al. (2017) [48] applied k-means clustering to autofluorescence multispectral images of 10 maize stems of the same genotype and identified up to 17 classes with different autofluorescence profiles. Lopez-Marnet et al. (2022) [30] segmented maize colour images of internode cross-sections stained with FASGA into 40 classes according to the hue, saturation, value and localisation of each pixel.

The spatial heterogeneity of cell walls shown via autofluorescence imaging is due not only to variations in the relative content of lignins and hydroxycinnamic acids but also to their chemical structure, molecular environment and interactions with other compounds within the cell walls without disregarding the history of the samples (storage, preparation, etc.). Due to the work of Donaldson and coworkers [35,40,41], some information on lignin fluorescence in planta has been obtained. However, its fluorescence is affected due to the copresence of hydroxycinnamic acids and their differentiation after UV excitation is not well understood. We hypothesise that the UV-induced fluorescence of ferulic and para-coumaric acids enables them to be distinguished; however, this needs to be further investigated. Different approaches can be implemented. Collecting fluorescence spectra of isolated pure compounds or tissues with a known composition is a good starting point. Combining *in situ* information on cell wall composition obtained on the same material by other methods, such as vibrational microspectroscopy, histochemical or immunolabelling approaches, is another approach. Similarly, the variations observed for mutants enriched or depleted in hydroxycinnamic acid compounds can also be studied.

## 5. Conclusions

Autofluorescence macroscopy, which combines a large field of view with good resolution and medium throughput imaging with an unbiased image analysis method is a preferred approach for exploring the spatial heterogeneity of the cell wall in organ sections. It can be applied to a series of biological samples for screening purposes. It can also be used as a first step in a multiscale approach to consider where higher spatial resolution imaging could be carried out. In this way, FTIR or Raman imaging could be implemented in a highly relevant way, by selecting the most representative regions of the larger sample.

Large Principal Component Analysis is a method that is easy to implement. The method can be applied to any series of multi- and hyperspectral images as long as the Principal Component Analysis is relevant to the spectroscopy under consideration, in particular infrared, Raman and Maldi imaging. 

In the present work, all sections of the image series could be compared using the large PCA score distributions and the inbred line effect could be quantified. This approach is valid when the regions observed in the images are comparable in structure and organisation. These distributions could also be used to segment images, as it is conventionally carried out with greyscale histograms in image analysis. 

Finally, the whole proposed methodology is an easy tool that could be advantageously applied to study the composition of the cell walls in varying environments or to compare original biological material depleted or enriched in certain phenolic constituents in close collaboration with plant physiologists.

## Figures and Tables

**Figure 1 biomolecules-13-01104-f001:**
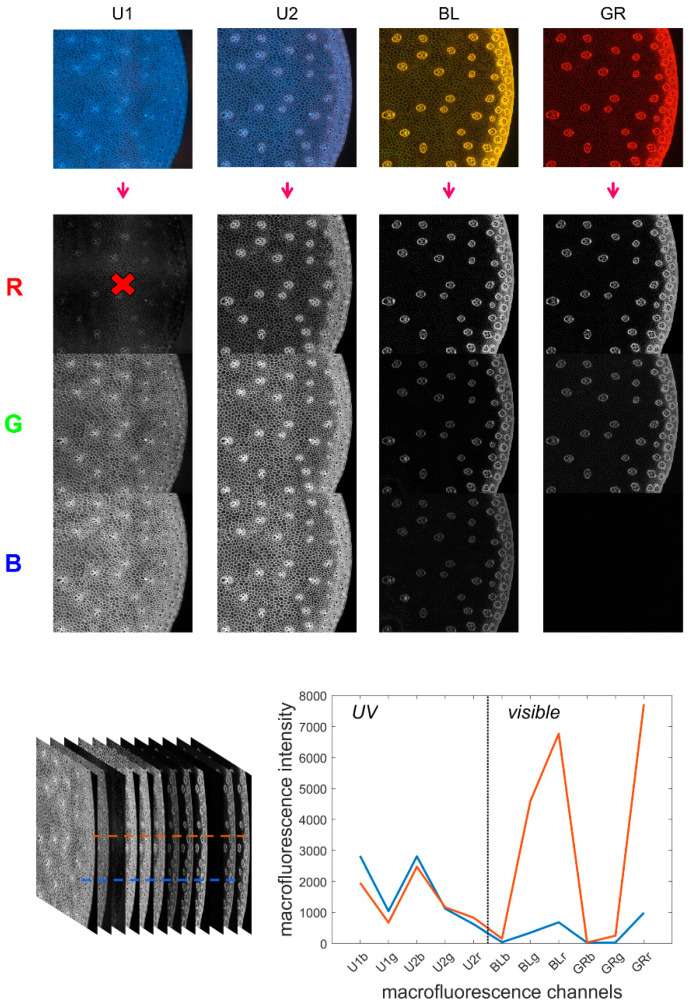
Multispectral image and pseudospectra. **Top**: from the four filter cube colour images to the 11 fluorescence channels. **Bottom**: the 11-channel autofluorescence multispectral image and two pseudospectra corresponding to individual pixels.

**Figure 2 biomolecules-13-01104-f002:**
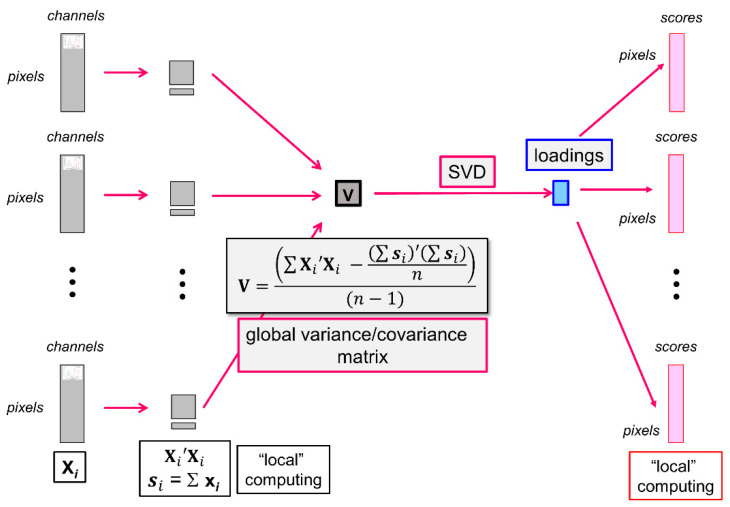
Principal Component Analysis of a series of large images. From left to right: unfolded images, local variables, global variance/covariance matrix, loadings, unfolded score images. SVD = Singular Value Decomposition.

**Figure 3 biomolecules-13-01104-f003:**
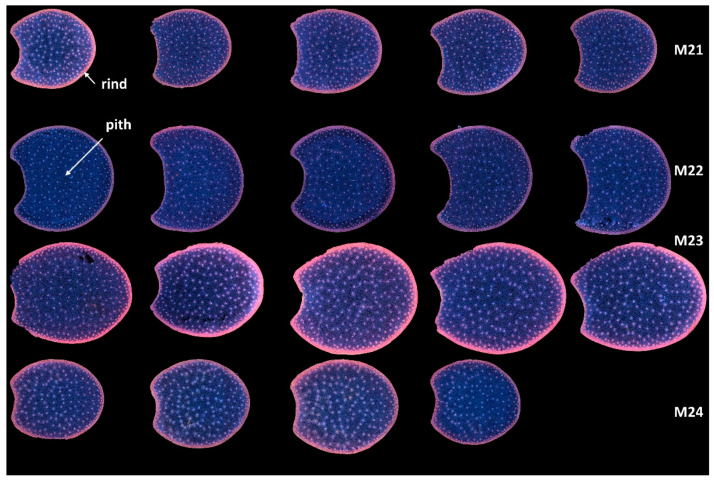
RGB composite images of macrofluorescence. Montage of one image per internode. The rind and the pith are the two main anatomical regions of the maize stem, with the rind surrounding the pith. Each row corresponds to one inbred line from top to bottom: M21, M22, M23 and M24. Each column corresponds to one internode. For M24, only four internodes are analysed. Colour intensities can be compared. The field of view of the whole montage is 7.65 cm × 11.45 cm, corresponding roughly to 2 × 2 cm^2^ per section.

**Figure 4 biomolecules-13-01104-f004:**
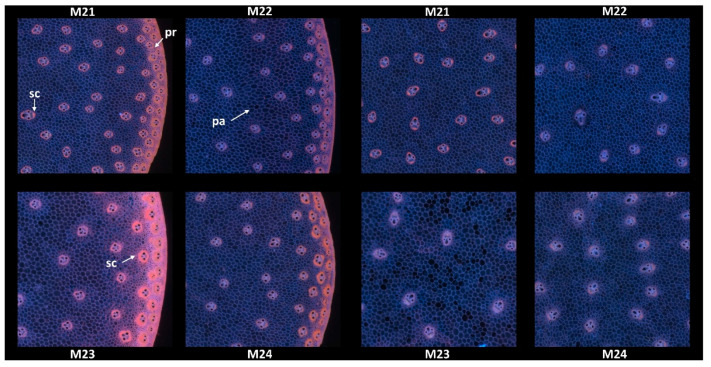
RGB composite images of macrofluorescence. Montages of one zoomed image per inbred line. Left: rind region and right: centre region; fields of view per image: 4.17 × 4.17 mm^2^. For the two montages, from top left to bottom right: inbred lines M21, M22, M23, M24. Colour intensities can be compared. Annotation legend: vb—vascular bundles, pa—pith parenchyma cell walls, sc—vascular bundles sclerenchyma sheath, pr—rind parenchyma.

**Figure 5 biomolecules-13-01104-f005:**
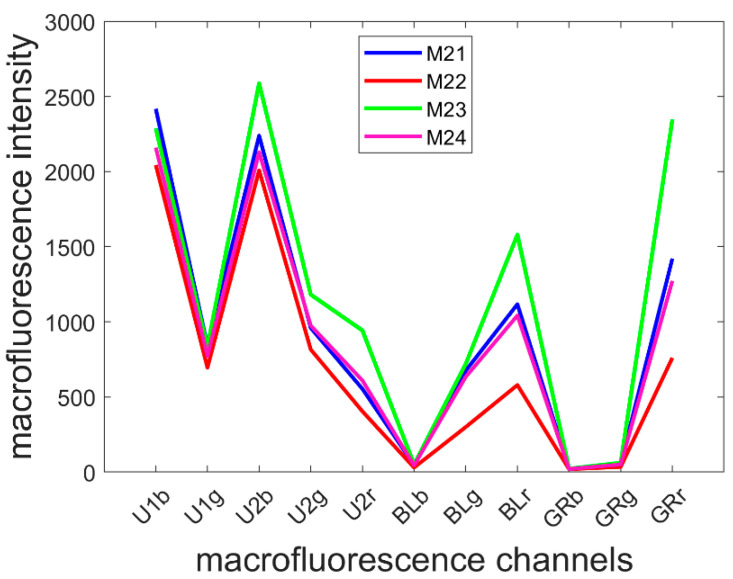
Average pseudospectra of the four inbred lines. Refer to Section 2.2.2 for the channel names. Channel U1r has been removed because it contained unwanted reflection.

**Figure 6 biomolecules-13-01104-f006:**
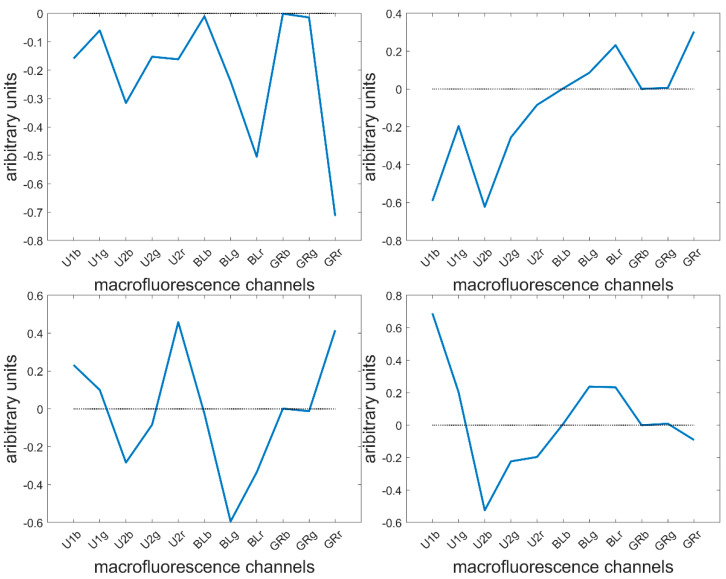
Large PCA of the entire series of 40 images. First four loadings.

**Figure 7 biomolecules-13-01104-f007:**
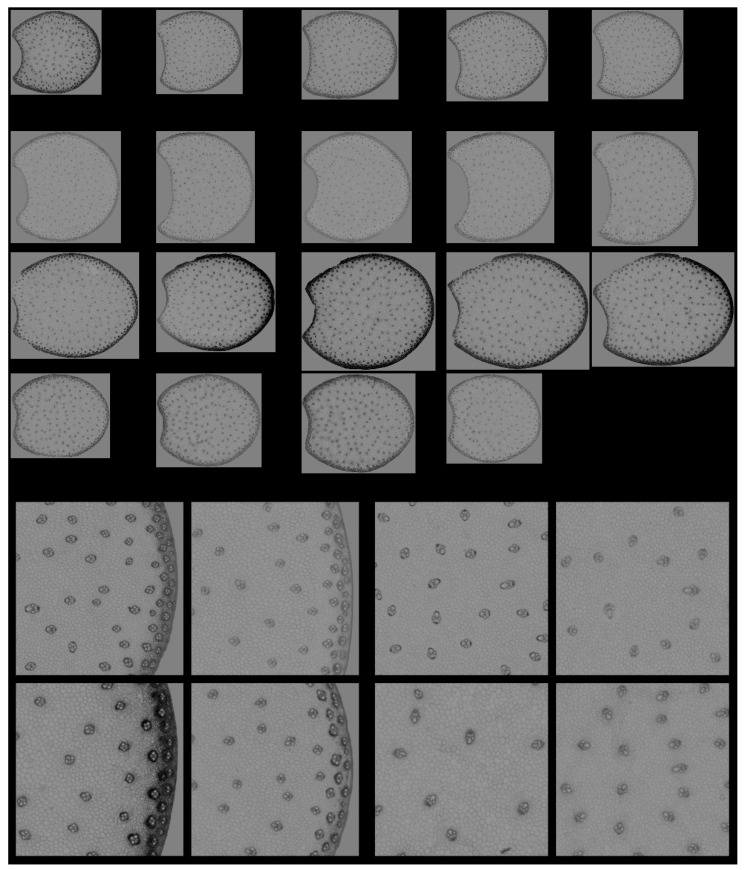
Large PCA. Principal Component 1 score images. **Top**: Montage of whole section images. Field of view: 7.65 cm × 11.45 cm. Each row corresponds to one inbred line: M21, M22, M23 and M24 from top to bottom and each column to one internode. **Bottom**: Montages of one zoomed image per inbred line. Left: rind region and right: centre region; fields of view per image: 4.17 × 4.17 mm^2^. For the two montages, from top left to bottom right: inbred lines M21, M22, M23, M24. Intensities can be compared.

**Figure 8 biomolecules-13-01104-f008:**
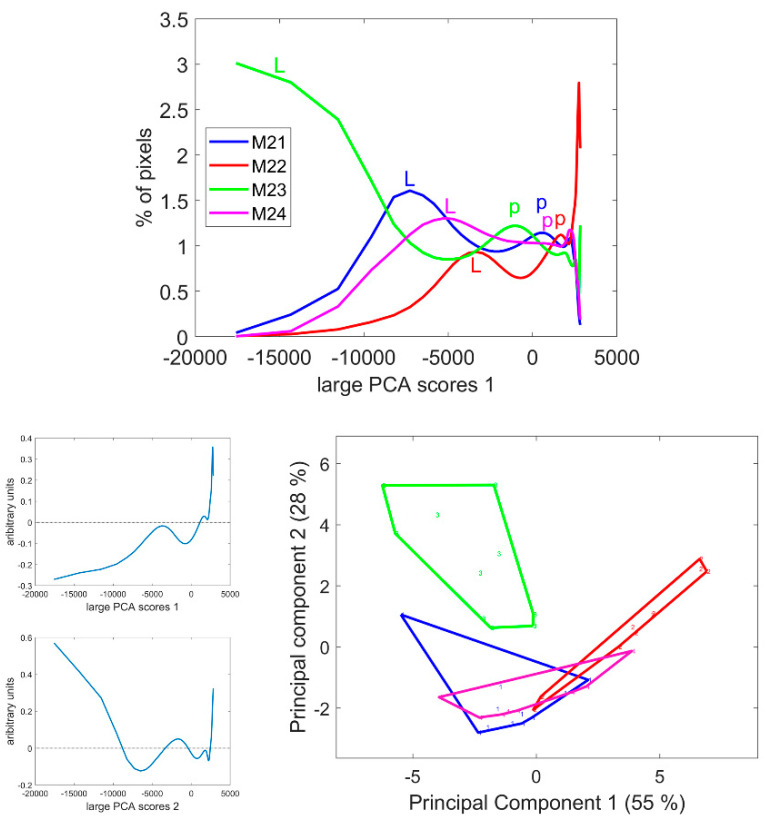
Score distributions of large PCA Component 1. **Top**: Average score distributions of the four inbred lines. **Bottom**: loadings and scores 1 and 2 of the Principal Component Analysis of the score distributions.

**Figure 9 biomolecules-13-01104-f009:**
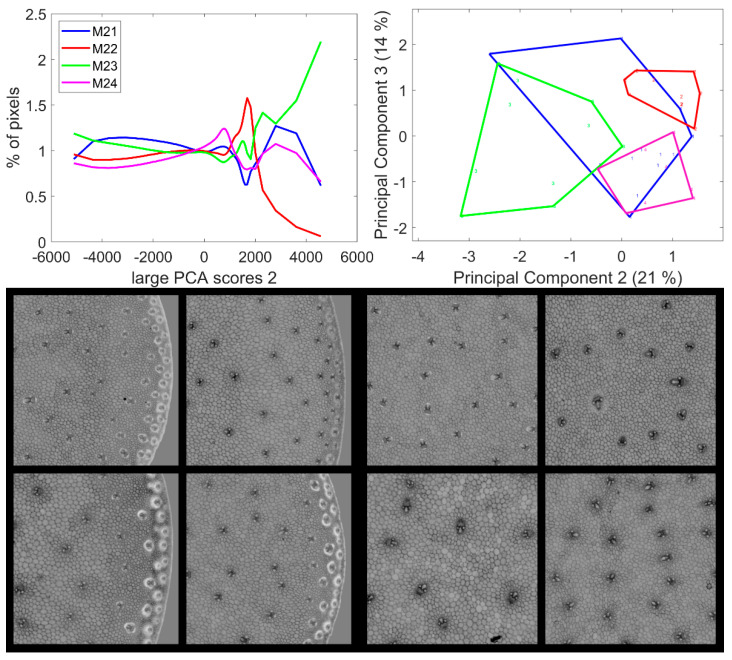
Large PCA. Principal Component 2 score images. **Top**: average score distributions and their Principal Components 2 and 3 (21, 14% of the total variance). Loadings can be found in Appendix A. **Bottom**: montages of one zoomed image per inbred line. Left: rind region and right: centre region; fields of view per image: 4.17 × 4.17 mm^2^. For the two montages, from top left to bottom right: inbred lines M21, M22, M23, M24. Intensities can be compared.

**Figure 10 biomolecules-13-01104-f010:**
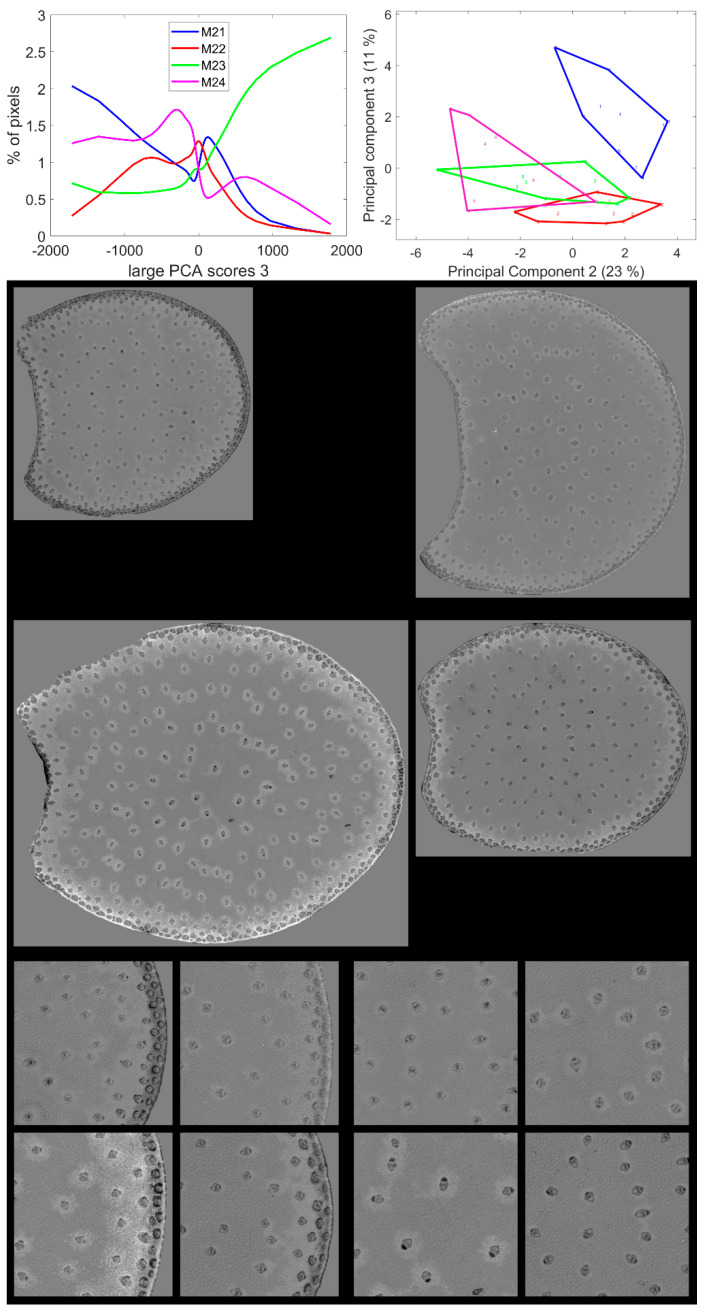
Large PCA. Principal Component 3 score images. **Top**: average score distributions and their Principal Components 2 and 3 (23, 11% of the total variance). Loadings can be found in Appendix A. Middle and bottom: montages of one image per inbred line. **Middle**: whole sections; total field of view: 3.82 cm × 4.60 cm. **Bottom**: left: rind region and right: centre region; fields of view per image: 4.17 × 4.17 mm^2^. For the three montages, from top left to bottom right: lines M21, M22, M23, M24. Intensities can be compared.

**Figure 11 biomolecules-13-01104-f011:**
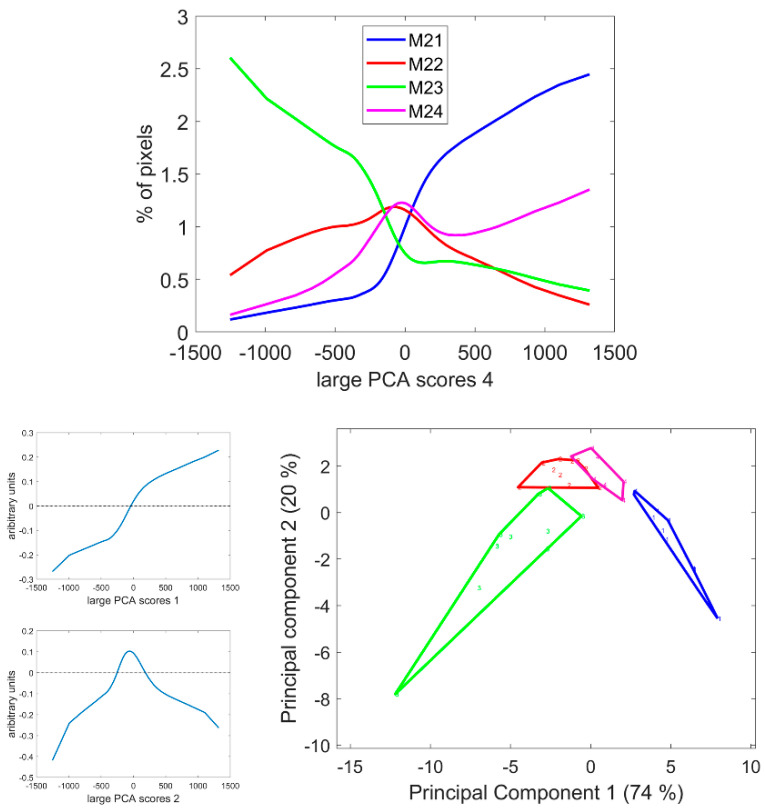
Score distributions of large PCA scores 4. **Top**: Average score distributions of the four inbred lines. **Bottom**: loadings and scores 1 and 2 of the Principal Component Analysis of the score distributions.

**Figure 12 biomolecules-13-01104-f012:**
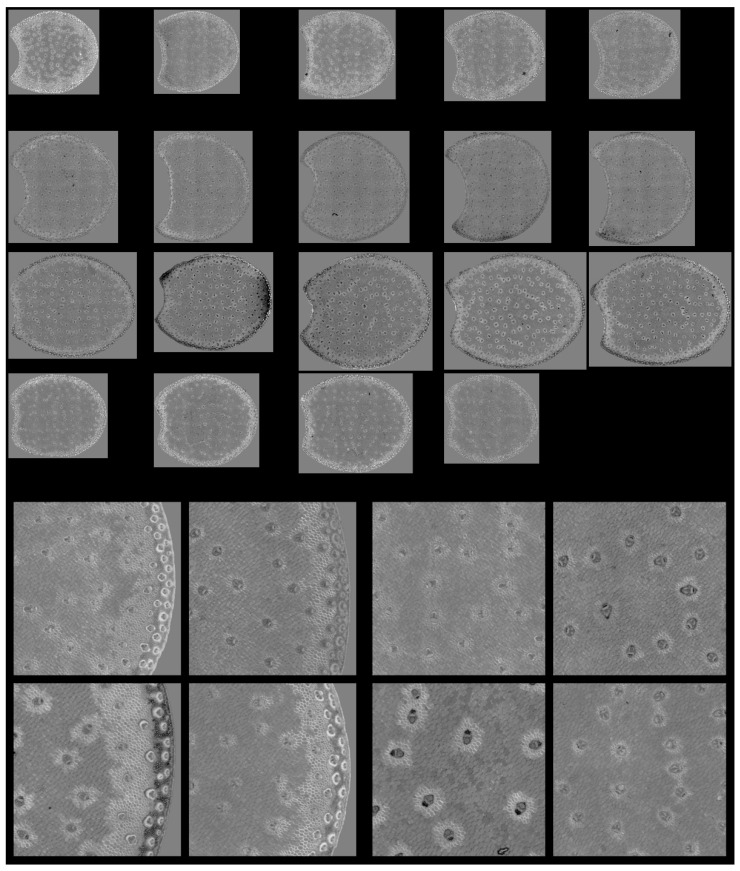
Large PCA. Principal Component 4 score images. **Top**: Montage of whole section images. Field of view: 7.65 cm × 11.45 cm. Each row corresponds to one inbred line: M21, M22, M33 and M24 from top to bottom and each column to one internode. **Bottom**: Montages of one zoomed image per inbred line. Left: rind region and right: centre region; fields of view per image: 4.17 × 4.17 mm^2^. For the two montages, from top left to bottom right: inbred lines M21, M22, M23, M24. Intensities can be compared.

**Table 1 biomolecules-13-01104-t001:** Characteristics of the four fluorescence filter cubes, acquisition time and gain.

Filter Code	Bandpass Excitation Filter (nm)	Dichroic Mirror (nm)	Longpass Emission Filter (nm)	Acquisition Time (ms)	Gain
U1	327–353	>380	>364	1500	10
U2	325–375	>400	>420	375	3
BL	460–490	>500	>515	250	3
GR	510–560	>565	>590	375	3

**Table 2 biomolecules-13-01104-t002:** Significance of the score distributions in describing the line effect. Percentage of the variance described by each Principal Components of the score distributions and *probability values (in italics)* of the variance analysis of the line effect.

Large PCA Scores	Score Distribution Principal Component 1	Score Distribution Principal Component 2	Score Distribution Principal Component 3	Score Distribution Principal Component 4	Score Distribution Principal Component 5
PC 1	54.5% *<0.000*	27.5% *<0.000*	5.8% *0.6111*	4.7% *0.0792*	2.6% *0.0862*
PC 2	54.2% *0.0954*	20.9% *<0.000*	14.3% *0.0015*	3.3% *0.0894*	2.4% *0.0021*
PC 3	57.8% *0.0001*	23.1% *<0.000*	11.2% *<0.000*	3.9% *0.0519*	2.0% *0.0350*
PC 4	74.4% *<0.000*	19.9% *<0.000*	3.7% *0.0940*	1.1% *0.0069*	0.5% *0.4450*

## Data Availability

The data presented in this study are available on request from the corresponding author.

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
