# Peer review of "Maize Internode Autofluorescence at the Macroscopic Scale: Image Representation and Principal Component Analysis of a Series of Large Multispectral Images"

_biomolecules, 2023, doi:10.3390/biom13071104_

Round 1

Reviewer 1 Report

Line 154, did not the author use microscope slides?

Line 161, I guess this should be a camera with a 14-bit dynamics that corresponds to 16384 grey-levels

Line 190, Specify: is this the average intensity per pixel of the whole image?

Line 192, actually, the B component with BL excitation does not appear negligible.

Figure 3, Add letters in each row to better identify images

Figure 3 caption, M23 instead of M33

Figure 4, Not clear correspondance between images and lines. Add letters in each panel to better identify images

Lines 289-292, Not easy to quantify difference by eyes

Lines 399-400, No statitics on Fig. 5 data are reported

Lines 404-407, From Fig.5 it appears that BLg is the same for M21, M23 and M24. BLr is higher for M23 followed by M21 and M24. It is not what reported in the text.

All data from PCA of images must be reviewed by an expert in this field.

Author Response

Reviewer 1 :

Thank you for the careful reading of our manuscript.

Line 154, did not the author use microscope slides?

            We haven’t used a slide+coverslip but we have used two round microscope cover glasses. It is now specified.

Line 161, I guess this should be a camera with a 14-bit dynamics that corresponds to 16384 grey-levels

Yes. Thank you, the number of grey levels has been updated and the dynamic range mentioned.

Line 190, Specify: is this the average intensity per pixel of the whole image?

            In the multispectral image, each pixel corresponds to 11 fluorescence intensities, each corresponding to the fluorescence intensity measured in one of the channels. This set of fluorescence spectral measurements can be considered as a spectrum. It is called a pseudo-spectrum to distinguish it from usual spectra. The objective of paragraph “pseudo spectra” was to define was is a pseudo spectrum. In fact, the paragraph was modified as suggestion by reviewer 4. We hope that it is clearer now.

Line 192, actually, the B component with BL excitation does not appear negligible.

Yes you are right. The emission filter of the BLUE fluorescence cube filter probably lets through some of the blue excitation light. However, the intensity is very low, as shown by the pseudospectra.

Figure 3, Add letters in each row to better identify images

The caption of the figure has been slightly modified to help the readers and the name of the inbred lines has been added on this figure. We prefer not to add any grids or names to the other montages in the series in Figures 7 and 12, as we feel that this would overload the figure and make it more difficult to observe the variations in grey levels in the images. In addition, the image annotation is made on powerpoint, which we are not fully sure of the way figures can be exported as image files. We hope that the modifications done help.

Figure 3 caption, M23 instead of M33

Done. Thank you.

Figure 4, Not clear correspondance between images and lines. Add letters in each panel to better identify images

Letters have been added on this figure but not on Figures 7,9,10,12 for the reason given in the answer for Figure 3.

The caption of the figure and that on Figures 7,9,10,12 have been modified. For Figure 4:

Figure 4: RGB composite images of macrofluorescence. Montages of one zoomed image per in-bred line: left: rind region and right: centre region; fields of view per image: 4.17*4.17 mm². For the two montages, from top left to bottom right: inbred lines M21, M22, M23, M24. Colour intensities can be compared. Annotation legend: vb: vascular bundles, pa: pith parenchyma cell walls, sc: vascular bundles sclerenchyma sheath, pr: rind parenchyma.

Lines 289-292, Not easy to quantify difference by eyes

We do not understand the comment if it concerns lines 289-292: ‘The minimum and maximum values were a priori determined from the eigenvalue l(i) of the large PCA component. The eigenvalues were used as an estimate of the standard deviation. The minima were set to -5*l(i) and the maxima to 5*l(i).’

We can imagine that it could have been for lines 389-392? ‘The zoomed images also showed that the relative number of vascular bundles varied; they were more numerous in M21. Their lignification was also probably different, as evidenced by the fluorescence of the sclerenchyma sheaths, which was more intense and orange in Line M21.’

We answer for these last lines and apologise if we have made a mistake

It is true that it is difficult to draw conclusions from images alone, and that the differences can be hard to appreciate. That is why we develop quantification tools such as large PCA to quantify and compare fluorescence emission of different samples.

We propose to modify ‘the fluorescence of the sclerenchyma sheaths, which was more intense and orange in Line M21’ by the fluorescence of the sclerenchyma sheaths, which seemed more intense and orange in Line M21.

Lines 399-400, No statitics on Fig. 5 data are reported

We did not put any statistic like standard deviations on the average pseudo spectra as it would have made the reading difficult. The statistics have been performed as indicated lines 407-408: ‘. Based on the mean pseudospectra, a line effect can be observed via Principal Component Analysis for example. The results are not shown here…’.

We propose to show the results of principal component analysis of the average pseudo spectra as supplementary materials. Sentences lines 407-408 have been modified to :’ Based on the mean pseudospectra, a line effect can be observed via Principal Component Analysis. The results are shown in Figure Sxxx but is not commented further here since this analysis cannot differentiate the various causes:  

Lines 404-407, From Fig.5 it appears that BLg is the same for M21, M23 and M24. BLr is higher for M23 followed by M21 and M24. It is not what reported in the text.

We apologise for not being clearer. The text describes the relative variations in fluorescence emitted in green and in red after excitation in blue. The ratio of the green over the blue emissions were respectively 0.60, 0.52, 0.46 and 0.60 for M21, M22, M23 and M24, respectively.

The sentence ‘Relative variations between green and red emissions after BLUE excitation were observed; more green emission (BLg) was found for Lines M21 and M24 than that for Lines M22 and M23.’ has been modified to : ‘Relative variations between green and red emissions after BLUE excitation were observed; compared to the red emission, a relative higher green emission (BLg) was observed for Lines M21 and M24 (ratio BLg/BLr of 060 ± 0.02 and 0.60 ± 0.04, respectively) than that for Lines M22 and M23 (ratio of 0.52 ± 0.05 and 0.46 ± 0.04, respectively).

All data from PCA of images must be reviewed by an expert in this field.

Reviewer 2 Report

The authors of this study have put a lot of efforts to study the quantitative histology of maize stalks in their end-use for quality. Maize internode autofluorescence at the macroscopic scale: image representation and Principal Component Analysis of a series of large multispectral images is a good Topic, yet it is needed to be modified with precise and brief concept of study. This study presented the problem of analyzing the large amount of data together by proposing the two-step approach. The study elaborated the problem in good manners. Figures are of good quality. Over all study was quite good, however following improvements must be made before further processing to improve the manuscript. English proficiency is highly recommended to be checked again by authors.

I recommend the manuscript after major revision.

1.      Manuscript needs a thorough revision for English proficiency.

2.      After first line in abstract, authors need to write the motive of study.

3.      Novelty in the abstract is missing.

4.      Future prospect should be mentioned clearly before ending the abstract.

5.      The introduction needs a proper story line. The link between paragraphs does not seem satisfactory.

6.      Line 65: It is better to indicate the duration of years rather just mentioning “For a few years”. It is quite ambiguous.

7.      Line 716: The conclusive remarks should remain separate.

8.      Future prospective should be mentioned after conclusion.

1Manuscript needs a thorough revision for English proficiency.

Author Response

Reviewer 2

The authors of this study have put a lot of efforts to study the quantitative histology of maize stalks in their end-use for quality. Maize internode autofluorescence at the macroscopic scale: image representation and Principal Component Analysis of a series of large multispectral images is a good Topic, yet it is needed to be modified with precise and brief concept of study. This study presented the problem of analyzing the large amount of data together by proposing the two-step approach. The study elaborated the problem in good manners. Figures are of good quality. Over all study was quite good, however following improvements must be made before further processing to improve the manuscript. English proficiency is highly recommended to be checked again by authors.

I recommend the manuscript after major revision.

First of all, we would like to thank the reviewer for its relevant comments, which helped us to improve the text.

  1. Manuscript needs a thorough revision for English proficiency.

We had the manuscript proofread for english before sending it out (The AJE Team www.aje.com. Order ID: 75P3ZJLQ). We have proofread carefully before sending back the revised version.

  1. After first line in abstract, authors need to write the motive of study.

This has been done. We add the following sentence “In the present work, a new methodology is proposed to show and quantify the spatial variability of tissue composition in plant organs and to statistically compare different samples accounting for biological variability.”.

  1. Novelty in the abstract is missing.
  2. Future prospect should be mentioned clearly before ending the abstract.

We have revised the abstract. The prospects for this study have been added at the end of the abstract and we hope that we have better highlighted the value of the whole approach.

  1. The introduction needs a proper story line. The link between paragraphs does not seem satisfactory.

We have revised the introduction. We have shortened certain paragraphs, particularly those concerning the interest of quantitative histology, and we have reworked the links between paragraphs. On reading, it seems to us that the introduction is much more fluid.

  1. Line 65: It is better to indicate the duration of years rather just mentioning “For a few years”. It is quite ambiguous.

The sentence has been changed. For the years, readers can refer directly to the publications

“In the case of maize stem, microscope slide scanners or macroscopes have been proposed to acquire images of whole stem sections [2, 3]”.

Line 716: The conclusive remarks should remain separate.

The paragraph title is ambiguous. In fact, this paragraph is part of the discussion, so we've changed the title “Maize stem tissues can be differentiated by their aufluorescence properties”

  1. Future prospective should be mentioned after conclusion.

A brief conclusion has been added, as well as some prospects

Reviewer 3 Report

The manuscript titled "Maize internode autofluorescence at the macroscopic scale: image representation and Principal Component Analysis of a series of large multispectral images" is well-written and effectively presents the authors' findings.

It requires further proofreading to correct minor errors.

Additionally, the abstract should be expanded. It is necessary to add new findings about new approach (from lines 745-750);

The word “stalk” should be substituted with a word “stem”.

It requires further proofreading to correct minor errors.

Author Response

Reviewer 3

The manuscript titled "Maize internode autofluorescence at the macroscopic scale: image representation and Principal Component Analysis of a series of large multispectral images" is well-written and effectively presents the authors' findings.

It requires further proofreading to correct minor errors.

First of all, we would like to thank the reviewer for its reading of our manuscript.

Additionally, the abstract should be expanded.

The abstract has been extended in response to this demand and comments made by reviewer 2.

It is necessary to add new findings about new approach (from lines 745-750);

A conclusion was added, highlighting the interest and limitations of the approach and proposing some perspectives for the next coming years.

The word “stalk” should be substituted with a word “stem”.

Done

Author Response

Reviewer 4 :

Thank you for the careful reading of our manuscript and for all suggestions to improve the clarity. Most of them have been retained.

57 : In the sentence "Therefore, the study and quantification of the heterogeneity and variability of tissue organisation and chemical composition is important," consider using the plural form "are" instead of "is" to match the subject "the study and quantification."

Done

82 : In the sentence "Autofluorescence imaging requires little tissue preparation, and macroscopic multispectral imaging is available [2, 42]," consider rephrasing the beginning of the sentence for clarity. For example, "The use of autofluorescence imaging requires minimal tissue preparation, and it is complemented by the availability of macroscopic multispectral imaging [2, 3]."

The sentence has been modified to: “It has the advantage of requiring little tissue preparation and there are imaging devices adapted to observe samples on a macroscopic scale [2, 3].”

94: In the sentence "The chemometric approach is generally considered, and the image is viewed as a set of spectra [33, 44, 45]," consider specifying which image is being referred to. For example, "In this context, the image being referred to is viewed as a set of spectra [33, 44, 45]."

The sentence has been modified to: In this framework, the multi or hyperspectral images are analysed as a set of spectra".

101: In the sentence "With the aim of comparing different samples, methods dedicated to the analysis of a series of images have been developed [2, 43, 46]," consider providing a brief description of these methods to give the reader a better understanding of what is being referred to. For example, "Methods such as [method names] have been developed with the aim of comparing different samples by analyzing a series of images [2, 43, 46]."

The sentence and following have been modified to: In order to analyse a series of images, different approaches have been proposed. In the study by Berger et al (2021) [2], a reduction in the number of spectra has been carried out in the following way. The 34 images of the series were individually segmented on the basis of morphological information and mean autofluorescence spectra were calculated for each segmented region. The mean spectra were then compared using Principal Component Analysis. In the Multivariate Curve Resolution framework, the analysis of several images is called multiset analysis [46, 47]. The authors developed the concept of data augmentation which consists of grouping together in a single table all the spectra from all images. In the study by Ghaffari et al. (2019) [43], 40 images were analysed together in this way, but the large number of pixels in the whole image series did not enable the consideration of the entire series in a single pass. In the study by Ghaffari et al. (2019) [43], each image was 512 x 512 pixels, resulting in more than 10 million pixels in the series, and in the study by Berger et al. (2021) [2], the 34 images were large images of approximately 4000x4000 pixels.

120 : In the sentence "Here, we propose a two-step approach," consider clarifying who the "we" refers to. For example, "In this study, a two-step approach is proposed."

The sentence has been modified to: “A two-step approach is proposed”

138 : Just the node? Or apportion of stem of X cm including the node

The sentence has been modified to: Five plants per inbred line were harvested at the silage stage, and only the internode located under the main ear was sampled

148 : How long? Days, weeks, or months

Sections stayed one year in alcool. It is now specified.

158 : What about this version of the paragraph? The paragraph is well-written in its current form, but

it may be clearer in the revised form below:

The multispectral autofluorescence images were acquired using the Multizoom AZ100M macroscope (Nikon, Japan). The macroscope was equipped with a Q Imaging EXI Aqua monochrome camera and an RGB-HM-S-IR filter wheel, which allowed for RGB color image acquisition and grey-level intensities coded using 16386 values. Excitation light was generated using an INTENSILIGHT mercury lamp (C-HGFI/C-HGFIE Precentred Fibre Illuminator Nikon, Japan). Four fluorescence filter cubes were used: UV1 (U1), UV2 (U2), BLUE (BL), and GREEN (GR), each with specific spectral specifications outlined in Table 1. To subtract the background, shading correction was applied to each image during acquisition. The total magnification was set to X4 by combining the AZ-Plan Fluor 2X lens (NA: 0.2/WD: 45 mm) and an X2 optical zoom. With these settings, each image represented a field of view measuring 3.9 x 2.9 mm² with a pixel size of 2.78 μm. The macroscope featured a Prior Proscan II (Nikon, Japan) motorized stage, enabling large image acquisition. The NIS-Elements software (AR 5.02.02) with the NDacquisition module was used to drive the macroscope for automated acquisition of large multispectral images. The multispectral sequence involved acquiring the four RGB images of the fluorescence filters in succession for a given field of view, starting with GREEN, followed by BLUE, UV2, and UV1 filter cubes. Exposure times were adjusted after viewing a few samples, as indicated in Table 1.

Thank you very much for the help in improving the paragraph. We have retained the suggestion.

182 : would suggest briefly summarizing the key points of publication 42 (34 in the revised version).

Publication [34] was the first publication of our group where we have proposed to analyse the autofluorescence multispectral images of the macroscope. We both cite the reference and describe again the way to consider and name the multispectral image channels.

The sentence has been modified to: “The multispectral images contained 12 channels by merging the four RGB images as proposed in [34]. The procedure is reminded here.”

190 : What about this version?

The collection of 11 fluorescence intensities measured for individual pixels was referred to as pseudospectra [34]. As no photons were emitted at wavelengths higher than the excitation wavelength, the signals in channels BLb, GRb, and GRr were absent for the visible filters (BLUE and GREEN). However, these channels were retained in the pseudospectra as a baseline reference. The pseudospectra can be divided into two regions: UV-induced fluorescence and visible-induced fluorescence, corresponding to the fluorescence obtained after UV excitation and BLUE/GREEN excitations, respectively.

Thank you very much for the help in improving the paragraph. We have retained the suggestion.

209 What do you think about removing this sentence or incorporating the expressed concept into

the "introduction" section?

“The representation of multispectral images is not a simple matter, and the visual comparison of the sets of large images requires adapted representation tools.”

210: The way the images were compared (even in the initial phase) is not clearly understandable. Please provide a clearer explanation of the concepts expressed.

We have modified the paragraph as follows:

Looking at images is only possible for monochrome and colour images. Multispectral images contain more than 3 channels and require appropriate representation tools to visualise them before they can be subjected to multivariate analysis. In this work, RGB composite images of macrofluorescence were calculated. Comparing several images creates additional problems. Common min and max values are necessary to compare intensities between images. Finally, in the case of large images, it is difficult to have a global view of several images and to be able to describe variations on the scale of details or regions in the images. In this work, automatic zoom region selections and image montages were set up to compare image series.

217 : I would suggest briefly summarizing the key points of publication 2, or are the following sentences explaining the key points of publication 2?

Yes, the following sentences explain the key points of publication [2].

244 Is the “Otsu threshold [50]” a function present in mathlab?

Yes. The name of the function is now given.

288 : Why just 8 bit?

Image scores were represented with a 8-bit precision as it is sufficient for screen or office display. Of course, such precision is not enough to perform additional image analysis such as segmentation. For this last purpose, images are also saved on disk as float in single precision.

Paragraph was modified as: After large PCA, scores were saved on disk in single precision. A representation has to be set up for score display and to compare score images from one image to the other. For this purpose, score intensities were coded to 8 bits, considering a common minimum and maximum for all images.

341 : Specify in the caption what is “Rind” and “Pith”

The sentence “The rind and the pith are the two main anatomical regions of the maize stem, with the rind surrounding the pith.” Has been added.

341: I would suggest placing a grid overlay on the image to indicate to the reader that the rows correspond to the genotypes M11, M22, M33, and M24, while the columns represent the different internodes. Are the genotypes arranged in any specific order, such as from lowest to highest? If so state in the caption.

No the genotypes are not ordered, excepted by their name. The caption of the figure has been slightly modified to help the readers and the name of the inbred lines has been added on this figure. We prefer not to add any grids or names to the other montages in the series in Figures 7 and 12, as we feel that this would overload the figure and make it more difficult to observe the variations in grey levels in the images. In addition, the image annotation is made on powerpoint, which we are not fully sure of the way figures can be exported as image files. We hope that the modifications done help.

381: Specify in the caption what is sc, vb pa pr

The annotation legends was already in the caption: “Annotation legend: vb: vascular bundles, pa: pith parenchyma cell walls, sc: vascular bundles sclerenchyma sheath, pr: rind parenchyma.”

359 : I would suggest exploring the correlation between blue fluorescence and hydroxycinnamic acids, referring to the cited publication.

Thank you for the suggestion. We indeed could have explored the relation between the global amount of hydroxycinnamic acids with the blue fluorescence intensity as in [2]. However, it was not the purpose of this work and we preferably envision to correlated spatial information using a complementary technique rather than global information.

394 : Please cite only the way used in this paper (calculate the mean pseudospectra of each multispectral image)

The first sentence of the paragraph was removed and the second one modified.

415 : I would suggest reminding the reader that U1r was not included.

Done.

418 Create a figure “Height components were computed describing 80.7, 16.1, 1.4, 1.0, 0.4, 0.2, 0.1 and 0.1% of the total variance”

We did not choose to make a figure for the percentage of total variance for several reasons. We don't base our decision on the percentage of variances to decide whether or not to look at the components. In particular in the case of images, a small percentage of pixels may have a specific behaviour that will not be reflected by a significant amount in the total variance. It can be the case for some of the tissues encountered in the vascular bundles.

437 : The resolution of the figure 6 seems too low

Yes, it is true on the pdf version of the manuscript but not on the word version. We have sent the figure in 300 dpi, which is normally sufficient. If not, we will generate the figure again with a higher dpi.

432 : On what basis? The two UV filters, although they were close in terms of excitation and emission, were hypothesised to be able to distinguish between ferulic and para-coumaric acids

The filters have close excitation/emission specifications: Exc: 327-353(dichroic) 380 em> 364 for the UV1 filter and exc 325-375 (dichroic) > 400 em> 420 for the UV2. It has been shown (Saadi et al, 1998) that two acids differ in their fluorescence properties after 365 nm excitation with a higher relative emission below 420 nm in the case of ferulic acids compared to the para coumaric acid. We have added the reference to the table and the paper and a sentence to refer to paragraph 3.7 where this point is more discussed. The full spectral interpretation would require more comment, but in our view this is beyond the scope of this document.

482 : Put a grid over the image or name the sub-images

See answer point 341.

496 : Put the values cited in the text as axis of PC1 e PC2 (54.5 and 27.5%)

We have replaced the values in the text as there are not specific needs for the precision and to keep the figure as clear as possible.

500 : I would suggest to use the approach of the function SCREE PLOT in R to select the number of dimensions to consider (Elbow Method).

Thank you for the suggestion. But as explain in point 418 over, we don’t base our choice of the number of components solely on the % variance. Ion the present work, we have considered the ability of components to describe the inbred line effect. In fac,t the fifth component showed also a line effect. We have chosen to show up to 4 components as the loadings and the score images revealed regions that could be ascribed to ferulic/paracoumaric variations.

579 Where does this sentence come from? Ferulic and para coumaric acids were found in maize stems.

We have added references [4,6, 8]

580 : And this? At neutral pH, a more specific blue fluorescence emission could be observed for ferulic acid and greener emission in the case of para-coumaric acid. The differences between the UV fluorescence of lignin and hydroxycinnamic acids were also expected.

References [55] and [33, 35] were added.

635 This is not the right place for this sentence. It should be placed at the end of the introduction.

Done. The sentence has been put at the beginning of the last paragraph of the introduction.

645 Add citations “Multispectral autofluorescence imaging can be used to assess the phenolic composition of the plant cell wall”

(655) : references have been added and the sentences has been modified to : “Multispectral autofluorescence imaging can be used to assess the distribution of phenolic compounds of the plant cell wall [33, 35].”  

724 : If this sentence ("Other authors demonstrated this diversity in cell wall composition in maize stem sections.") is referring to the following bibliographic citations (from row 725 to 729), it is not accurate because they only demonstrate a difference in spectra. Only Donaldson and coworkers [35, 39, 40], demonstrated differences in lignin even if affected by hydroxycinnamic

The sentence has been modified to: “Other authors have also found a diversity in cell wall local composition depending on tissue and localisation in maize stem sections.”

737 : The sentence "We hypothesize that the UV-induced fluorescence of ferulic and paracoumaric acids enables them to be distinguished; though, this needs to be further investigated" is the most significant in the entire work. The text presents a relatively easy method to investigate where plants exhibit differences, but it does not attribute these differences to specific causes. I believe that a close collaboration with plant physiologists and the group that wrote this work could yield excellent results.

We would like to thanks again the reviewer for the comments and suggestion for improving our manuscript.
